# Clock Genes, Inflammation and the Immune System—Implications for Diabetes, Obesity and Neurodegenerative Diseases

**DOI:** 10.3390/ijms21249743

**Published:** 2020-12-21

**Authors:** Elaine Vieira, Gerardo Gabriel Mirizio, Geovana Reichert Barin, Rosângela Vieira de Andrade, Nidah Fawzi Said Nimer, Lucia La Sala

**Affiliations:** 1Postgraduate Program on Physical Education, Universidade Católica de Brasília, DF, Taguatinga 71966-700, Brazil; 2Muscle Cell Physiology Laboratory, Center of Molecular Studies of the Cell, Institute of Biomedical Sciences, Faculty of Medicine, Universidad de Chile, 8330015 Santiago, Chile; gerardo.mirizio@gmail.com; 3Postgraduate Program in Genomic Science and Biotechnology, Universidade Católica de Brasília, DF, Taguatinga 71966-700, Brazil; geovanareichert@gmail.com (G.R.B.); rosangelavand@gmail.com (R.V.d.A.); nidahsaid@hotmail.com (N.F.S.N.); 4Laboratory of Cardiovascular and Dysmetabolic diseases, IRCCS MultiMedica, 20138 Milan, Italy; lucia.lasala@multimedica.it

**Keywords:** inflammation, neurodegenerative diseases, clock genes, diabetes, obesity

## Abstract

Inflammation is a common feature of several diseases, including obesity, diabetes and neurodegenerative disorders. Circadian clock genes are expressed and oscillate in many cell types such as macrophages, neurons and pancreatic β cells. During inflammation, these endogenous clocks control the temporal gating of cytokine production, the antioxidant response, chemokine attraction and insulin secretion, among other processes. Deletion of clock genes in macrophages or brain-resident cells induces a higher production of inflammatory cytokines and chemokines, and this is often accompanied by an increased oxidative stress. In the context of obesity and diabetes, a high-fat diet disrupts the function of clock genes in macrophages and in pancreatic β cells, contributing to inflammation and systemic insulin resistance. Recently, it has been shown that the administration of natural and synthetic ligands or pharmacological enhancers of the circadian clock function can selectively regulate the production and release of pro-inflammatory cytokines and improve the metabolic function in vitro and in vivo. Thus, a better understanding of the circadian regulation of the immune system could have important implications for the management of metabolic and neurodegenerative diseases.

## 1. Introduction

The activity of the immune system is coordinated by intrinsic molecular clocks in blood T and B lymphocytes, monocytes, macrophages and other inflammatory cells. Circadian rhythms within the immune system are synchronized by a ‘master’ clock located in the suprachiasmatic nuclei (SCN) of the hypothalamus with a timing of ~24 h. At a molecular level, the core clock system of the SCN produces rhythms of gene expression by means of self-sustained transcriptional/translational feedback loops [1]. The main loop is formed by BMAL1 and CLOCK transcription factors, which heterodimerize in the nucleus driving the transcription of Period *(Per1,2)* and Cryptochrome *(Cry1,2)* genes. PERs and CRYs accumulate in the cytoplasm, after which they enter the nucleus and inhibit BMAL1/CLOCK, producing rhythmic oscillations in gene expression. Further regulation is driven by the nuclear receptors REV-ERBs (*α* and *β*) and RORs (*α*, *β*, and *γ*) that repress and activate BMAL1, respectively. Importantly, the localization and stability of circadian clock proteins is regulated by some members of the family of casein kinases (CKI) and protein phosphatases (PP) [2]. In addition to the core clock system in the SCN, peripheral clocks have been found in the heart, skeletal muscle, liver, kidney, pancreas and immune cells [3,4,5,6,7,8,9]. In immune cells, the activation of inflammatory signals is mediated by clock genes [10]. The levels of expression of clock genes in immune cells are affected in pathologies characterized by chronic and systemic inflammation such as neurodegeneration, obesity and diabetes. Recent studies have pointed out the importance of the circadian gene *Rev-erbα* (reverse erythroblastosis virus alpha) as a mediator between inflammation and circadian rhythmicity. Altogether, this highlights the importance of the molecular clocks in mediating inflammatory responses and their potential role as therapeutic targets for the treatment of neurodegenerative pathologies. This mini-review discusses the recent findings regarding the role of clock genes in neuroinflammation, obesity and diabetes and new strategies for the treatment of neurodegenerative diseases.

## 2. Clock Genes, Neuroinflammation and the Immune System

Circadian clock genes have been implicated in the regulation of several diseases where inflammation constitutes a hallmark. Neuroinflammation, for example, is a common feature of many neurodegenerative diseases such as diabetes, Alzheimer’s, Parkinson’s and Huntington’s disease. Circadian clocks and sleep can influence the key processes involved in neurodegeneration, and this has led to consideration of the circadian systems as potential targets to promote brain health [11]. To address the role of circadian clocks in neurodegeneration, Musiek et al. [12] generated *NestinCre^+^:Bmal1^flox/flox^* mice in which *Bmal1* (brain and muscle ARNT-like 1, also known as *Arntl*) is deleted in most neurons, astrocytes and oligodendrocytes, with residual *Bmal1* expression in microglia. *Bmal1* knock-out animals showed an increased expression of proinflammatory cytokines in the cortex, which is consistent with a condition of chronic neuroinflammation. These animals also presented behavioral abnormalities and dysregulation in the oxidative stress response. Strikingly, deletion of *Bmal1* in the cortex also caused an 83% decrease in the expression of *Rev-erbα*, another circadian clock component, which suggested a possible mechanism by which *Bmal1* could regulate neuroinflammation. Indeed, the role of *Rev-erbα* in neuroinflammation was later confirmed in a *Rev-erbα^−/−^* mouse model. Griffin et al. [13] observed that *Rev-erbα* deletion in mice caused spontaneous microglial activation in the hippocampus and promoted a proinflammatory phenotype. This was revealed by an up-regulation of the NF-κB pathway, a critical regulator of innate immune system activation. Furthermore, a physical interaction between REV-ERBα and several NFκB–related genes was confirmed by chromatin immunoprecipitation (ChIP). Remarkably, the same study demonstrated that the REV-ERBα agonist SR9009 suppressed brain inflammation in vivo, suggesting that the activation of *Rev-erbα* could constitute a viable strategy for the treatment of neurodegenerative pathologies.

It has been proposed that the regulation of inflammation by the immune system may underlie the pathologies associated with circadian desynchrony. The relationship between the immune system and circadian rhythms has been evaluated in several transgenic mice models. Castanon-Cervantes et al. [14] imposed a severe challenge to the immune system of *Per2^Luc^* knock-in mice by means of high doses of *E. coli* endotoxin lipopolysaccharides (LPS), injected intraperitoneally. In this study, a group of animals was exposed to chronic circadian disruption with a weekly 6-h phase advance over 4 weeks. As a result, this group of animals exhibited a magnified response to LPS compared to mice that was not exposed to chronic circadian disruption. This was reflected by higher levels of pro-inflammatory (i.e., IL-1β, GM-CSF, IL-12, and IL-13) and lower levels of anti-inflammatory (i.e., IL-10) cytokines in the serum of the phase-shifted mice group. Furthermore, macrophages harvested from the phase-shifted mice group exhibited an enhanced response to LPS in vitro and a significant decrease in *Bmal1* expression. Thus, it was proposed that the dysregulation of inflammatory responses by circadian disruption could be mediated, at least partially, by an alteration in the clock function of peritoneal macrophages. In order to test the role of the macrophages’ clock genes in the temporal gating of cytokine responses, Gibbs et al. [15] generated a macrophage-specific *Bmal1−/−* mice (*LysMbmal−/−),* which carried a luciferase reporter for the circadian clockwork (mPER2:Luc). Macrophage-specific *Bmal1* knock-out mice exhibited a suppression of transcripts for the clock gene *Rev-erbα*, in agreement with the observations by Musiek et al. [12] in the brain. Then, with the aim of testing the role of *Rev-erbα* in the gating response of cytokines in vivo, they assessed the effect of the endotoxin LPS-induced IL-6 response in *Rev-erb^α−/−^* mice. As a result, the rhythmic immune responses were abolished in the serum and in macrophages from these animals (Figure 1). Furthermore, it was observed that REV-ERBα has control over several genes involved in innate immunity, including *Il6, Il19, Cxcl6, Cxcl11,* and *Ccl2*. Recently, Sato et al. [10] showed that REV-ERBα mediates the inflammatory infiltration of macrophages by repressing the expression of *Ccl2*. CCL2 is an important chemokine that binds the CCR2 receptors in monocytes/macrophages to stimulate their migration and initiate inflammation, CCL2 promotes cell adhesion and migration by activating the ERK- and p38-signaling pathways, which are also targeted by REV-ERBα. Moreover, the REV-ERBα agonist GSK4112 inhibited the expression of *Ccl2* following LPS stimulation. Altogether, these studies demonstrate that REV-ERBα constitutes a key link between the cellular circadian timers and innate immune responses.

Apart from REV-ERBα, the clock gene *Bmal1* also controls the circadian response of the innate immune system. The role of *Bmal1* was investigated in myeloid cells, by studying its role in controlling the rhythmic trafficking of Ly6C^hi^ inflammatory monocytes. Ly6C^hi^ monocytes provide the first line of defense against the gram-positive bacterium *Listeria monocytogenes*. In this sense, Nguyen et al. [16] showed that the specific deletion of *Bmal1* in myeloid cells induced a higher serum concentration of inflammatory cytokines and chemokines, such as IL-6, IL-1β, IFNγ and CCL2. This was accompanied by a disruption in the circadian patterns of Ly6C^hi^ monocytes and a higher susceptibility of mice to infection-induced systemic inflammation. In addition to its role in mediating the inflammatory response, *Bmal1* also plays a crucial role in promoting the antioxidant response in myeloid cells [17]. Mice with *Bmal1* excised in the myeloid lineage exhibit a decrease in the relative activity of the NRF2 transcription factor, leading to an accumulation of reactive oxygen species (ROS), activation of the HIF1A transcription factor and a higher production of the proinflammatory cytokine IL-1β. Thus, BMAL1 limits the production of the proinflammatory cytokine IL-1β by targeting the oxidative stress pathways in macrophages. In brain-resident cells, *Bmal1* also plays a role in controlling the inflammatory responses. In microglia from Per1:Luc transgenic mice, *Bmail1* knock-out selectively inhibited LPS-induced IL-6 expression [18]. Then, ChIP assays demonstrated that *Bmal1* binds an E-box element in the promoter region of the *Il6* gene. Finally, to assess the role of *Bmal1* in vivo, these authors generated mice with *Bmal1* deficiency in CD11b cells (*Bmal1^−/−^_CD11b_*). After exposure to middle artery cerebral occlusion to cause a severe inflammatory event in the brain, *Bmal1^−/−^_CD11b_* mice exhibited a less potent up-regulation of IL-6. Altogether, these results suggest that the clock gene *Bmal1* modulates the inflammatory response by up-regulating IL-6 expression in the brain.

Apart from the role of clock genes in controlling the inflammatory response, pro-inflammatory agents can also affect circadian rhythmicity. Experimental data suggest that TNFα and IL-1β can inhibit the activation of E-box regulatory elements of clock genes by CLOCK/BMAL1 [19]. Furthermore, the effects of TNFα on clock gene expression occur specifically in the SCN, and lead to prolonged rest episodes during spontaneous activity of mice. This effect of TNFα on locomotor activity may provide a link between the immune system activation and fatigue associated with immune diseases. In addition to TNFα, other inflammatory agents can also affect the circadian clock function. Among them, IFNγ affects the circadian rhythms and gene expression in SCN neurons [20]. This effect is accompanied by a decrease in spontaneous excitatory postsynaptic and spiking activity of SCN neurons. Similarly, LPS produces a transient suppression of circadian clock genes and clock-controlled genes in the SCN of rats [21]. Although not assessed in this study, it has been suggested that there is an indirect action of LPS through inflammatory cytokines such as IL- 6, TNFα and IFNγ. Thus, a better understanding of the bidirectional relationship between clock genes and inflammatory signals could give us new insights of the pathogenesis of inflammatory diseases.

## 3. Clock Genes, Inflammation, Obesity and Diabetes

Chronic inflammation plays a role in the development and progression of several metabolic diseases, including obesity and diabetes. In the context of such pathologies, the chronic activation of pro-inflammatory signaling pathways leads to a low-grade inflammation condition and, in many cases, to insulin resistance [22,23]. At the same time, a link has been established between metabolic diseases and circadian misalignments [24]. Since inflammation is a common feature of obesity and diabetes, it has been suggested that it plays a critical role in mediating the circadian disruption in metabolic diseases.

In the context of obesity, there is increasing evidence that immune cell plasticity and versatility play a key role in the development of the disease. The phenotype switching of macrophages from the anti-inflammatory M2-type to the pro-inflammatory M1-type is important for the initiation and amplification of adipose tissue inflammation [25]. During obesity, macrophage proinflammatory activation aggravates the inflammation in peripheral tissues such as the liver and the adipose tissue, thereby leading to systemic insulin resistance. The relationship between circadian clock dysregulation and macrophage proinflammatory activation in diet-induced obesity was evaluated in a *Per1^ldc^/Per2^ldc^* mutant mice, characterized by disruption of the clock genes *Per1* and *Per2* [26]. Bone marrow-derived macrophages (BMDM) from *Per1/2* knock-out mice displayed a significantly higher percentage of proinflammatory M1-type macrophages. Furthermore, the macrophage proinflammatory activation in response to LPS was higher in mutant mice, as reflected by an increased JNK1 and NF-κB p65 phosphorylation, and a higher expression of the proinflammatory cytokines IL-1B and TNFα. In addition, the transfer of BMDM from *Per1/2* knock-out mice into wild-type mice potentiated inflammation and systemic insulin resistance. Although it is unknown how *Per1* and *Per2* controls macrophage activation in obesity, the authors suggested a disruption in the feedback regulation of CLOCK/BMAL1 protein complex. Alternatively, this may occur through the interaction of *Per1/2* with the REV-ERBα protein, which modulates macrophage TLR signaling. The PPARγ transcription factor may also play a role in *Per1/2* regulation of macrophage activation, since its levels decreased in conjunction with an increased proinflammatory state. Overall, this study shows that high-fat diet disrupts clock genes function in macrophages, inducing a proinflammatory activation of such cells and contributing to systemic inflammation and insulin resistance.

Diabetes mellitus (DM) is a group of metabolic diseases characterized by hyperglycemia as a result of insulin secretion/action defects [27]. The disruption of clock genes is also involved in the development of DM, and this is especially relevant in pancreatic β cells. Intrinsic molecular clocks in β cells are responsible for the regulation of glucose-stimulated insulin secretion, β cell maturation/turnover and the response to diabetogenic stressors. Proinflammatory cytokines are examples of diabetogenic stressors that, in contact with β cells, play a prominent role in the development and progression of Type 1 (T1DM) and Type 2 (T2DM) diabetes mellitus [28,29]. In T1DM, β cells are exposed to the proinflammatory cytokines IL-1β, TNFα and IFNγ; however, in T2DM the proinflammatory state is characterized by the interplay between IL-1β and IL-6 cytokines circulating in the adipose tissue, IL-1β production in islet-derived macrophages and cytokine production in islet cells. To address the role of circadian rhythms in pancreatic β function during the development of diabetes, islets from Per2:Luc-MIP:GFP reporter mice were incubated for 72 h with the pro-inflammatory cytokines IL-1β, TNFα, IL-6, and IFNγ [30]. As a result, a dampening of the amplitude and phase of *Per2*-driven luciferase oscillations was observed. The cytokine with the most deleterious effect on the β cell circadian clocks was IL-1β, which disrupted them by altering the expression of the NAD^+^ dependent deacetylase SIRT1. SIRT1 is a critical regulator of the expression, stability and overall function of the *Bmal1* clock gene. Strikingly, the treatment of β cells with the SIRT1 activator Resveratrol restored *Bmal1* gene expression and β cell dysfunction after exposure to IL-1β. To address the role of circadian transcriptional regulators in the development of diabetes, pancreatic tissue from patients with T2DM was collected and analyzed by immunofluorescence. Immunoreactivity of BMAL1, SIRT1 and RORα was attenuated in β cell from patients with T2DM as compared to a control group. These observations suggest a link between islet inflammation, circadian clock disruption and β cell failure in T2DM. In agreement with this, Rakshit and Matveyenko [31] observed a protective role of BMAL1 in β cells dysfunction. By means of a mouse model in which BMAL1 is overexpressed specifically in β cells (β-*Bmal1*^OV^), they analyzed the effects of BMAL1 on glucose-stimulated insulin secretion (GSIS), islet transcriptomic and glucose metabolism in a diet-induced obesity context. BMAL1 overexpression in mice fed with a high-fat diet enhanced in vivo glucose tolerance, β cell function and expression of genes involved in insulin secretion, endoplasmic reticulum function and lipid metabolism. Finally, a Per2:LUC reporter mice expressing green fluorescent protein (GFP) under the control of the insulin promoter (*Per2^luc/+^Ins1^GFP/+^)* was used to assess the effect of a pharmacological treatment on β cell clock function. The administration of the polymethoxylated flavone Nobiletin, a potent clock amplitude-enhancing compound, enhanced the circadian clock amplitude of β cells and GSIS in transgenic mice. Thus, it is tempting to speculate that therapeutic strategies directed to enhance circadian clock function could improve the pathological condition in T2DM patients.

Finally, it is important to determine whether the circadian clock dysregulation is linked to the fatty acid content of the diet, which is a key determinant of inflammatory signals and insulin resistance in metabolic diseases. The effect of different high-fat diets on circadian gene expression has been studied in human skeletal muscle biopsies from obese, insulin-resistant men [32]. Muscle samples were collected prior to and 4 h after the consumption of a high fat-containing meal consisting of either saturated (SFAs), monounsaturated (MUFAs) or polyunsaturated (PUFAs) fatty acids. Gene expression was assessed by microarray technology, enabling changes to be identified in several genes involved in circadian rhythms regulation, inflammation and the antioxidant/stress response. High-fat meal consumption damped the expression of the clock genes *Per1, Per3, Rev-erba* and *Rev-erbβ*, among other clock-controlled genes. However, the SFAs caused more prominent changes in circadian clock gene expression than MUFAs or PUFAs containing meals. The authors proposed that high-SFA intake negatively affects the muscle function by altering the circadian clock gene expression and leading to inflammation and oxidative stress in humans. A recent study conducted by Kim et al. [33] assessed the role of proinflammatory cytokines in the modulation of circadian clock gene expression response triggered by SFAs. Using *Bmal1-dLuc* fibroblast cultures, this study found that the treatment with the proinflammatory cytokines IL-6 or TNFα induced phase advances of *Bmal1* dependent rhythms. Strikingly, these effects were suppressed when neutralizing antibodies against IL-6 or TNFα were utilized. This role of inflammatory cytokines as key mediators of SFA-induced circadian desynchrony may also have important therapeutic implications for the treatment of metabolic diseases.

## 4. Conclusions

Inflammation constitutes a hallmark of many age- and obesity-related diseases, such as neurodegenerative diseases, metabolic syndrome, obesity and diabetes. Many cells involved in inflammatory responses, such as macrophages, neurons and pancreatic β cells, possess an intrinsic molecular clock and show circadian rhythms of gene expression. In the context of inflammation, these endogenous clocks control the temporal gating of cytokine production, the antioxidant response, chemokine attraction and insulin secretion, among other processes. Strikingly, deletion of clock genes in mice causes a dysregulation of inflammatory, oxidant and insulin secretion responses. This is reflected by dysregulation of pro- (e.g., IL-1β, IL-6, IL-12, TNFα) and anti-inflammatory (e.g., IL-10) cytokines, as well as key metabolic enzymes (e.g., SIRT1) and transcriptional factors (e.g., NF-κB, NRF2, HIF1A). Remarkably, pharmacological activation of clock genes can suppress the inflammatory and oxidative phenotype in brain and immune cells. Therefore, it has been suggested that targeting clock genes could constitute a viable strategy for the treatment of inflammatory diseases.

## Figures and Tables

**Figure 1 ijms-21-09743-f001:**
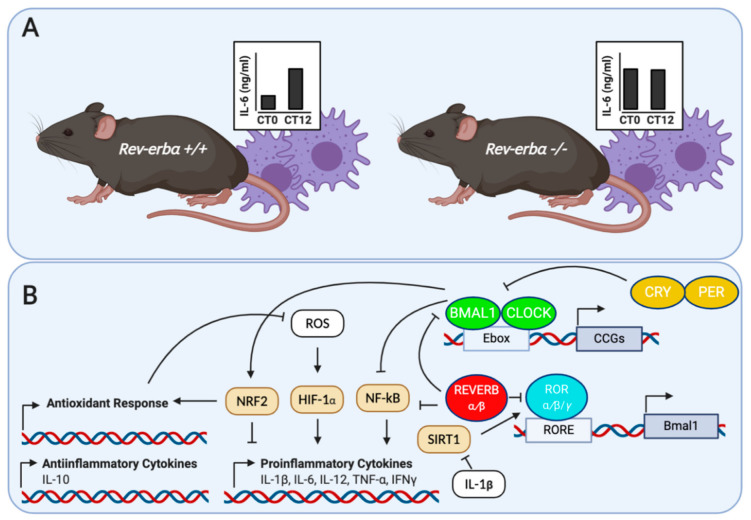
(**A**) Loss of temporal gating of IL-6 cytokine response in serum and macrophages from *Rev-erb^+/+^* and *Rev-erb^α−/−^* in response to LPS. Adapted from Gibbs et al. [15]; (**B**) molecular regulation of cytokines and oxidant response in inflammatory cells by circadian clock proteins. BMAL1 and REV-ERBα control the inflammatory response by down-regulating NF-κB-mediated transcription in inflammatory cells. In addition, BMAL1 regulates the IL-1β response in macrophages via a NRF2-dependent suppression of ROS and HIF-1α. Finally, the pro-inflammatory cytokine IL-1β is able to disrupt β cell clocks by altering the expression of the NAD^+^ dependent deacetylase SIRT1.

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
