# Peer review of "Clock Genes, Inflammation and the Immune System—Implications for Diabetes, Obesity and Neurodegenerative Diseases"

_ijms, 2020, doi:10.3390/ijms21249743_

Round 1
Reviewer 1 Report
In this manuscript, Vieira and co-workers present an interesting and timely overview about the relationship between the circadian clock genes, inflammation and diseases in which inflammation is a strong mediator of disease, in particular, obesity, diabetes and neurodegenerative diseases. Overall, the manuscript is well-written and most of the references are up-to-date. Please consider the following comments.
1 - The title should be rearranged, is a bit misleading for readers. As suggestion: “Clock genes, inflammation and the immune system – implications for diabetes, obesity and neurodegenerative diseases”.
2 - The abstract should be improved; is not able to stand alone and does not make a presentation about the work that will be discussed and presented in the review.
3 – Keywords should include neuroinflammation and/or neurodegenerative diseases.
Author Response
In this manuscript, Vieira and co-workers present an interesting and timely overview about the relationship between the circadian clock genes, inflammation and diseases in which inflammation is a strong mediator of disease, in particular, obesity, diabetes and neurodegenerative diseases. Overall, the manuscript is well-written and most of the references are up-to-date. Please consider the following comments.
1 - The title should be rearranged, is a bit misleading for readers. As suggestion: “Clock genes, inflammation and the immune system – implications for diabetes, obesity and neurodegenerative diseases”.
Response: We have changed the titled as suggested. Thank you for the comment on this.
2 - The abstract should be improved; is not able to stand alone and does not make a presentation about the work that will be discussed and presented in the review.
Response: We re-wrote the Abstract according to the following structure: lines 16 to 20, inflammation and circadian clocks; lines 20 to 22, focus on macrophages and brain-resident cells; lines 22 to 24, focus on obesity and diabetes; lines 25 to 30, therapeutic implications.
3 – Keywords should include neuroinflammation and/or neurodegenerative diseases.
Response: The term “neurodegenerative diseases” was included as a key word. Since “inflammation” was already included as a key word, we decided not to include “neuroinflammation” to avoid a term redundancy.
Reviewer 2 Report
In this review by Vieira et al, authors describe the role of certain circadian clock-regulating genes in controlling diabetes through inflammation.
Comments:
Major Comments:
1) The review is highly superficial; In order to provide knowledge pertaining to the subject discussed, it needs to be more detailed, especially in describing diabetes and the link between circadian clock and diabetes.
2) For better understanding, please also extend on clock genes, neuroinflammation and the immune system
Minor Comments
1) Detailed figure legends will be helpful to understand figures better.
2) Please provide the extensions of certain abbreviations, such as Rev-erbα
Author Response
In this review by Vieira et al, authors describe the role of certain circadian clock-regulating genes in controlling diabetes through inflammation.
Major Comments:
1) The review is highly superficial; In order to provide knowledge pertaining to the subject discussed, it needs to be more detailed, especially in describing diabetes and the link between circadian clock and diabetes.
Response: We expanded on the section CLOCK GENES, INFLAMMATION, OBESITY AND DIABETES by incorporating more details to the studies. Also, we added two new studies to this section (ref.31 and 33). Please, note that a brief definition of Diabetes Mellitus was also incorporated to this section, just as requested.
2) For better understanding, please also extend on clock genes, neuroinflammation and the immune system
Response: We also expanded on this section by incorporating more details to the description of the studies. Also, we included a new paragraph in which we discuss the effect of inflammatory agents on circadian clock genes and rhythmicity.
Minor Comments
1) Detailed figure legends will be helpful to understand figures better.
Response: Thank you for the suggestion. We included a detailed explanation to Figure 1. Please, note that minor changes were also applied to Figure 1A so as to get a better picture of the Gibbs’ findings (see Gibbs et al. 2011 in ref.).
2) Please provide the extensions of certain abbreviations, such as Rev-erbα
Response: We included the full name of all the clock genes to the manuscript, as well as the full names of other abbreviations (e.g. ChIP, ROS, DM).